**Subject Category:**
Biology (whole organism)

behaviour

song behaviour, zebra finch, song variability

**Author for correspondence:**
Allison L. Lansverk
e-mail: alansverk@gmail.com

# The variability of song variability in zebra finch (*Taeniopygia guttata*) populations

Allison L. Lansverk[1], Katie M. Schroeder[1],
Sarah E. London[2], Simon C. Griffith[3], David F. Clayton[4]
and Christopher N. Balakrishnan[1]

[1]Department of Biology, East Carolina University, Greenville, NC, USA
[2]Department of Psychology, University of Chicago, Chicago, IL, USA
[3]Department of Biological Sciences, Macquarie University, Sydney, New South Wales, Australia
[4]Department of Biological & Experimental Psychology, Queen Mary University of London, London, UK

 ALL, 0000-0002-4311-6690; SEL, 0000-0002-7839-2644;
CNB, 0000-0002-0788-0659

Birdsong is a classic example of a learned social behaviour. Song behaviour is also influenced by genetic factors, and understanding the relative contributions of genetic and environmental influences remains a major goal. In this study, we take advantage of captive zebra finch populations to examine variation in a population-level song trait: song variability. Song variability is of particular interest in the context of individual recognition and in terms of the neuro-developmental mechanisms that generate song novelty. We find that the Australian zebra finch *Taeniopygia guttata castanotis* (*TGC*) maintains higher song diversity than the Timor zebra finch *T. g. guttata* (*TGG*) even after experimentally controlling for early life song exposure, suggesting a genetic basis to this trait. Although wild-derived *TGC* were intermediate in song variability between domesticated *TGC* populations and *TGG*, the difference between domesticated and wild *TGC* was not statistically significant. The observed variation in song behaviour among zebra finch populations represents a largely untapped opportunity for exploring the mechanisms of social behaviour.

## 1. Introduction

As a representative of the Oscine Passerines, or songbirds, zebra finches have been the subject of extensive neurobiological and behavioural research with a focus on vocal communication

behaviour [1–10]. Despite its role as a model system, however, zebra finch song behaviour is in some ways unusual among songbirds. For example, zebra finches are highly age-restricted or 'closed' learners that do not modify their songs after around 90 days of age [11]. Zebra finches also maintain high song structural diversity even within local populations, yet show little evidence for the regional song dialects well-known in songbirds [12,13].

As zebra finches are highly colonial birds, population-level song structural diversity may facilitate individual recognition. Exposure to novel song has been shown to cause differential behavioural, electrophysiological and gene expression responses relative to playbacks of songs to which birds have previously been exposed [14], and this recognition learning or habituation must be dependent in part on salient structural variation in song. Juvenile zebra finches regularly produce novel syllables that are distinctive from those of their tutors [15–17]. Lachlan *et al.* [12] suggested that the lack of local dialects may be the result of high variability in zebra finch song.

Might zebra finch song variability itself be a trait subject to variation, selection and ultimately genetic control? Comparison of existing zebra finch populations could provide insight into this. The majority of zebra finch research relies on domesticated populations of the *Taeniopygia guttata castanotis* (*TGC*) subspecies, native to Australia. Relative to many domesticated species, the domestication of the zebra finch has been recent (approx. 150 years) and closely related wild and wild-derived populations are still available for comparison [18–20]. Previous studies have demonstrated significant genetic differentiation between domesticated and wild populations and a loss of genetic diversity in captive populations [21]. There is also evidence for human-mediated selection on zebra finches including selection for larger body size [20,21]. Unlike many laboratory models, however, captive zebra finch populations are often kept as more natural, free-mating, outbred colonies. Thus, despite a loss of genetic diversity, there is no evidence of severe genetic bottlenecks [21]. Timor zebra finches *T. g. guttata* (*TGG*), native to the Lesser Sunda Islands of Southeast Asia, have more recently been established in the pet trade and are also beginning to see experimental research use [22–24]. The colonization of the Lesser Sunda Islands by zebra finches appears to have occurred about one million years ago and was accompanied by a severe population bottleneck [25]. Previous work has described differences between *TGC* and *TGG* in song length, frequency and the number of notes per song phrase [26]. In this study, we build on these known subspecific differences and test for population differences in song variability itself. Through experimental tutoring, we also attempt to determine whether differences in song copying fidelity might explain population differences in song variability.

# 2. Material and methods

## 2.1. Study populations

We recorded songs of individuals sampled from four populations of *TGC*: University of Illinois at Urbana-Champaign (UIUC), East Carolina University (ECU), University of Chicago (Chicago), and Macquarie University (Macquarie), and two populations of *TGG*: UIUC and ECU. We consider the UIUC, ECU, and Chicago populations of *TGC* to be domesticated, in that their origins trace to long-term captive populations in the USA. All of the zebra finches used in this study were bred, raised and housed in large flight aviaries and thus were exposed to similar acoustic environments and a rich social environment.

The ECU *TGC* colony was founded by five pairs of birds in 2012. These five pairs were drawn from another captive population housed at ECU (courtesy of Dr Ken Soderstrom) which itself was founded in 2002 with birds sourced from Acadiana Aviaries (Franklin, Louisiana, USA). When sampled, the ECU *TGC* colony included approximately 20 pairs of birds. *TGC* recorded at ECU were housed in the same room as *TGG*, but the two large aviaries were on opposite ends of the room, with a mixed species aviary (*Lonchura striata*, *Poephila acuticauda*) in between them. The Chicago population was derived from adult birds sourced from Magnolia Bird Farm (Anaheim, CA, USA). One Chicago colony was originally founded in October 2011 by a set of 40 males and 40 females. In June 2013 another colony of 50 males and 50 females was founded. Each of these two populations at Chicago bred independently until they were merged in November 2015. Birds recorded here were drawn from this large, merged colony. The *TGC* breeding colony at UIUC (recorded in 2009) was founded in 2002 with birds from Magnolia Bird Farm and maintained at a size of 30–50 breeding pairs with periodic additions of birds from various sources. The *TGC* colony at Macquarie was founded in 2007 using approximately 100 birds taken directly from the wild in Australia [27], with an additional 40 birds, also taken directly from the wild, added to the population in 2010. These wild-derived birds have been kept isolated from domesticated birds and have had the opportunity to breed about every 12 months.

The ancestors of the *TGG* finches used in this study were originally brought into captivity in the USA in the early 1990s (efinch.com, San Jose CA, USA). The UIUC *TGG* colony was founded by five pairs in 2009 and their descendants were moved to ECU in 2012 where a colony of approximately 25 pairs has been maintained. Thus, the *TGG* birds recorded at ECU were direct descendants of those at UIUC, separated by at least five generations of uncontrolled breeding in the aviary. The recording of these birds across generations allows us to test whether initially low levels of song diversity were caused by the small founding population size. Among the colonies sampled, parentage is only tracked in the Macquarie population. Birds were selected from aviaries for recording using haphazard sampling. For the Macquarie birds, however, we confirmed that recorded males were not derived from the same nest.

## 2.2. Song recordings

We first recorded songs of *TGC* ($n = 4$) and *TGG* ($n = 5$) in 2009 at UIUC. We then recorded another 10 individuals of each subspecies at ECU between 2014 and 2016. In order to put the findings of the subspecies comparison in a broader context, we recorded an additional domesticated *TGC* population at the University of Chicago ($n = 8$) and the wild-derived population at Macquarie University ($n = 7$). In all cases, pairs of birds (one male, one female) were placed in a sound chamber and recorded using the activity-triggered recording in Sound Analysis Pro 2011 (SAP2011 [28]). *TGG* rarely vocalized in the absence of a female, so this approach was required in order to collect comparable song data. Birds were left in the chamber until the males produced 100 song bouts, which were used in subsequent analyses.

Birds at ECU were recorded using a Sennheiser ME22 shotgun microphone connected to a Focusrite Scarlett 2i2 USB pre-amplifier, which was in turn connected to IBM Thinkpad laptop running SAP2011. The wild-derived colony at Macquarie University was recorded in July 2016 at Macquarie using the same equipment. Birds at the University of Chicago were recorded during April 2017 using a Rode NT5 Small-diaphragm Cardioid Condenser Microphone connected to Focusrite Scarlett 2i2 USB Pre-amplifiers; songs were recorded via Dell OptiPlex desktop computers running SAP2011.

By pairing a male and a female, we captured a mixture of female-directed and undirected song, the latter of which is known to be more variable [29]. In order to assess the relative proportion of directed song, another set of *TGC* ($n = 6$) and *TGG* ($n = 6$) at ECU were observed through a webcam (Logitech HD Pro C 920) positioned inside the sound chambers. Observations occurred between 09.00 and 12.00, generally lasting for 1 h but ranging from 0.5 to 2 h depending on the rate of vocal behaviour. At least 10 song bouts per male were categorized as either directed or undirected song. To distinguish between the two song types, we adapted Woolley & Doupe's [30] definition of directed song as bouts where the male oriented toward the female and additionally hopped, wiped his beak, fluffed his body feathers, or groomed the female.

## 2.3. Experimental tutoring

Population differences in song variability could be caused by differences in social environments or by subspecific differences in the ability to copy the songs of tutors (either through differences in tutor song memorization, sensorimotor error correction and/or motor ability). In order to understand the mechanisms underlying any differences in song variability, we conducted a song learning experiment with *TGC* and *TGG* housed at ECU. Individual male zebra finches were placed with a male Bengalese finch, *Lonchura striata domestica*, tutor in a small cage; these cages had visual and partial acoustic barriers between them. These cages were also located in the large room described above. Thus, the housing room also contained flight aviaries with both zebra finch subspecies and Bengalese finches, but tutoring cages were spatially separated from the aviaries to minimize social interactions between cages and aviaries. Nests and perches were located on the opposite side of the aviaries as the tutoring cages, maximizing the distance between the zebra finch colonies and the experimental birds. In total, five different Bengalese finch tutors were used, balanced among treatments as evenly as was feasible, with each Bengalese finch tutoring at least one individual of each subspecies. For logistical reasons we used two different strategies for tutoring, moving eggs to foster parents ($n = 3$ *TGC*, $n = 2$ *TGG*), and moving birds at post-hatch day 30 (p30) to tutoring cages ($n = 7$ *TGC*, $n = 6$ *TGG*). Our original method was to move eggs, but because of the low success rate of viable hatches as well as a high rate of tutors rejecting cross-fostered chicks we experienced, we switched to moving birds at p30 once the chicks became independent. Tutors were assigned based on the hatching time of the subjects and the availability of tutors (birds that were not currently tutoring other subjects). All individuals were left with their tutors until at least day 90, after the

close of the critical period for song learning and song crystallization [1,31–34]. After this period, the tutees and tutors were recorded in the sound chamber in the method described above.

## 2.4. Song analysis

Following recording, syllables shorter than 20 milliseconds were filtered to remove cage noise from recordings. We used the Feature Batch function in SAP to parse the motifs into syllables by manually setting segmentation values for amplitude, mean frequency and continuity once for each individual and then parsing all recordings. As a final noise removal step, we used the tclust R package a non-hierarchical method for robust clustering [35]. Recordings were trimmed by importing the measurements of syllable duration and 44 birds were filtered at an alpha value of 0.03. Older recordings taken at the UIUC were notably less noisy and were trimmed at an alpha of 0.01 in order to avoid filtering obviously repeated syllables.

To quantify song variability, we estimated Kullback–Leibler (K-L) distance using KLFromRecordingDays [36,37]. K-L distance is ideal for song repertoire comparisons in zebra finches because it mitigates the challenge of identifying homologous syllables in distinctive individual repertoires. K-L distance allows us to quantify the difference between sets of syllables by reducing the syllables to two-dimensional scatter plots for each set being compared and comparing the probability density function. A K-L of zero would indicate that the two sets being compared are identical, whereas a higher K-L distance indicates that the two patterns generated by the set of syllables are less similar, i.e. songs that are more dissimilar among the individuals being compared [36]. Because multiple syllables could be very similar in a two-dimensional song space, the number of clusters underestimates the number of unique syllables.

Each K-L distance is a measurement of the difference between two song repertoires. The average K-L distance between all pairs of birds in a population is therefore an index of population-level song variability. These population estimates of K-L divergence included comparisons in which each individual bird was used as both a 'template' and a 'target' to generate a full matrix of pairwise comparisons. Because K-L distance is an asymmetric measurement, we took the average of both comparisons using each pair of birds to generate an estimate of song similarity between each pair of birds (a half matrix of pairwise comparisons). We also used K-L distance to quantify similarity between experimentally tutored birds (*TGC* and *TGG*) and their tutors. K-L distances were calculated for 13 song parameters estimated by the batch analysis function in SAP2011 (amplitude, pitch, frequency modulation (FM), squared amplitude modulation (AM), Wiener entropy, pitch goodness, mean frequency, and the variance in pitch, FM, entropy, pitch goodness, mean frequency and AM).

## 2.5. Statistical analyses

K-L distance was estimated in pairwise comparisons of individuals within populations (*TGC*: Macquarie, Chicago, UIUC, ECU; *TGG*: UIUC, ECU), between tutored birds and their tutors, and within groups of experimentally tutored birds (*TGC* and *TGG*). All statistical analyses were performed using R [38]. For each comparison we used K-L distance as a response measure in Bayesian linear mixed models (*blmer* function in *blme package*) treating Population as a fixed factor. The *blmer* function applies a weak prior (Wishart) over the random effects to avoid singularity. In an overall test of population variability, we compared our four largest samples (*TGC*: ECU, Chicago, Macquarie; *TGG*: ECU). To account for repeated comparisons of individual birds among pairwise distance estimates, we treated both the 'template' and the 'target' as random factors nested within Population. We tested model assumptions (homoscedasticity of residuals) using residuals plots and q–q plots. Confidence intervals around estimated means were extracted using the *predict* function. Our primary analysis of variability in *TGG* and *TGC* subspecies included the recordings from ECU used above, but also the smaller set of birds sampled at UIUC. The inclusion of this additional sample allowed us to statistically control for batch effects (the university where the recordings took place) in recording. Analyses that compared song copying fidelity between tutors and their *TGG* and *TGC* tutees did not involve repeated comparisons among tutees. However, because tutors were used more than one time, we included the ID of the tutor as a random factor in these models.

All K-L distance estimates include syllable duration as one of the variables but were estimated for the remaining 13 secondary song parameters listed above. As K-L distances are all measures of song dissimilarity, K-L estimates based on individual song parameters were highly correlated. We described these correlations using principal components analysis (PCA) using the *prcomp* function on

log-transformed K-L distance estimates for each song parameter. We conducted separate PCA for unmanipulated birds ($n = 310$ pairwise distance estimates), among experimentally tutored birds ($n = 20$ pairwise distance estimates, 10 per subspecies) and between experimentally tutored birds and their tutors ($n = 18$ pairwise distance estimates, 10 for *TGC*, 8 for *TGG*). Each PCA was based on the 13 different parameters estimated by SAP2011.

Because syllable duration is incorporated into all K-L distance measures and *TGC* appeared to incorporate some longer syllables than *TGG*, we also directly compared syllable duration measures between *TGC* and *TGG* using ANOVA and the *aov* function in R. We compared the maximum syllable duration for each individual for both normally raised and experimentally tutored *TGG* and *TGC* at ECU.

# 3. Results

## 3.1. Overall patterns of song variability

In an overall PCA of all of the non-experimentally tutored birds, PC1 explained 86.5% of the variation in the song data. All of the song parameters had similar loadings on PC1 ranging from amplitude (0.23) to frequency modulation (0.30). Based on the correlation among measures of K-L distance we elected to use PC1 as the primary response measure in all of our statistical tests.

## 3.2. Song variation among subspecies

For a direct comparison of the two subspecies, we included *TGG* and *TGC* birds sampled at UIUC and ECU. This approach allowed us to explicitly test for batch effects among birds recorded at different sites, at different times and with different equipment. Including 'batch' in our models did not improve them ($\chi^2 = 0$, d.f. = 1, $p = 1$) so we removed batch from the model and proceeded to test for differences between subspecies. This analysis revealed a significant difference in song variability between subspecies for PC1 ($\chi^2 = 11.1$, d.f. = 3, $p = 0.01$) with *TGG* showing lower song variability than *TGC* (figure 1). Due to small sample size for the UIUC population we did not statistically compare *TGG* finches recorded at ECU with those from UIUC. Both samples, however, showed lower diversity relative to *TGC* (figure 1*a*). In a group of six *TGC* and six *TGG* finches video recorded subsequent to these song analyses, we observed no difference between subspecies in rates of directed versus undirected songs (*TGC* mean = 14.6% directed song (85.4% undirected), *TGG* mean = 13.2% directed song (86.8% undirected), two-tailed Mann–Whitney *U* test, $U = 17$, $z = 0.08$, $p = 0.94$) so the observed differences in song variability among subspecies cannot be explained by variation in the frequency of directed song.

## 3.3. Song variation after controlled tutoring

In general, song copying of Bengalese finches by zebra finches was relatively poor. We could not confidently identify matched syllables using spectrograms, so we again relied on K-L distance as an index of song similarity. Under high song copying fidelity, we would expect tutor and tutee syllables to match in acoustic space. Within the two-dimensional scatter plots shown in figure 1*c* for one highlighted feature (FM), distinct clusters for the Bengalese tutor (black) and its zebra finch tutees (blue for *TGC* and orange for *TGG*) are apparent. This was confirmed by the relatively high K-Ls for all the parameters between tutors and tutees. To summarize, K-L distances between tutored subjects and their tutors we conducted another PCA. In examining these K-L distances, PC1 explained 60% of the variation in the data and we again used PC1 as our measure of song variability. Loadings for each of the song parameters on PC1 ranged from 0.24 (amplitude) to 0.30 (mean frequency). There was no overall effect of timing of tutor exposure (egg versus p30) on how closely they matched their tutor in PC1 ($\chi^2 = 0.48$, $p = 0.49$). We found no difference between *TGC* and *TGG* in terms of how closely they matched their tutor songs ($\chi^2 = 1.41$, d.f. = 1, $p = 0.23$).

We also tested for differences in song variability among subspecies by comparing the average K-L distance within each subspecies after experimental tutoring. Again, we summarized these K-L distances using PC1, which in this case explained 72% of the variation in the data. Despite this lack of a statistical difference in song copying fidelity described above, we did observe a significant difference between subspecies on song variability ($\chi^2 = 14.1$, d.f. = 1, $p = 0.0001$; figure 1*b*). Thus, among birds recorded at ECU, *TGC* maintained higher song diversity than *TGG* even after controlling for developmental song exposure. One possible explanation for differences in song variability between subspecies is that normally raised *TGC* sing some longer syllables than *TGG* ($F = 30.04$, d.f. = 1, $p = 3.3 \times 10^{-5}$) and thus could

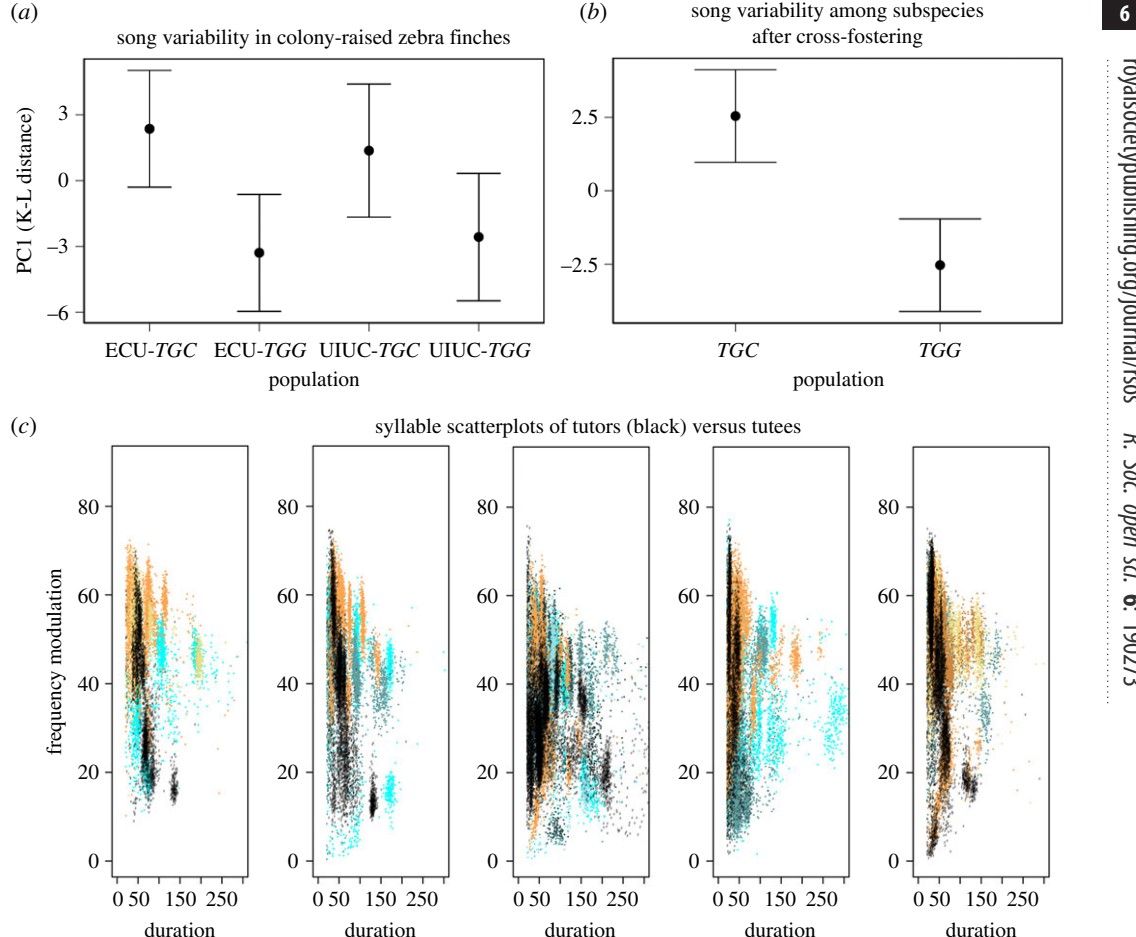

**Figure 1.** (a) Fitted means and confidence intervals for population estimates of K-L distance for principal component 1 for unmanipulated *T. guttata castanotis* (TGC) and *T. guttata guttata* (TGG) sampled at the University of Illinois (UIUC) and East Carolina University (ECU) ($\chi^2 = 11.1$, d.f. = 3, $p = 0.01$). (b) Fitted means and confidence intervals for population estimates of K-L distance for principal component 1 within populations of TGG and TGC after experimental tutoring by Bengalese finches ($\chi^2 = 14.1$, d.f. = 1, $p = 0.0001$). (c) Scatterplots of song repertoires of cross-fostered individuals and tutors. Distinct clusters reflect unique song syllables. Each of the five scatter plots represents one Bengalese tutor (black points) and its respective TGC (shades of blue) and TGG tutees (shades of orange).

simply be exploring additional acoustic space. After experimental tutoring, however, there was no difference between subspecies in average length of the longest syllable sung ($F = 0.50$, d.f. = 1, $p = 0.49$, figure 1c).

## 3.4. Comparison among additional populations of zebra finches

To test for differences in variability among populations we compared our four largest samples including two domesticated *TGC* (ECU, Chicago), one wild-derived *TGC* (Macquarie) and *TGG* (ECU). This analysis revealed a significant effect of population ($\chi^2 = 11.2$, d.f. = 3, $p = 0.01$; figure 2) with domesticated *TGC* showing relatively high song diversity, and *TGG*, as expected, showing low diversity. Interestingly, wild-derived populations were intermediate in song variability (figure 2). Given that *TGG* was clearly distinctive in its low variability, we further examined the three *TGC* populations (Chicago, ECU, Macquarie) to determine whether there was population variation within *TGC*. In this case, a model including Population as a fixed effect, however, was only marginally better than the null model ($\chi^2 = 2.6$, d.f. = 1, $p = 0.1$).

## 4. Discussion

Our analysis reveals differences in population-level song variability across zebra finch populations made up of the two zebra finch subspecies, one of which includes the long-term domesticated populations

(a)                     syllable scatterplots of colony-raised zebra finches

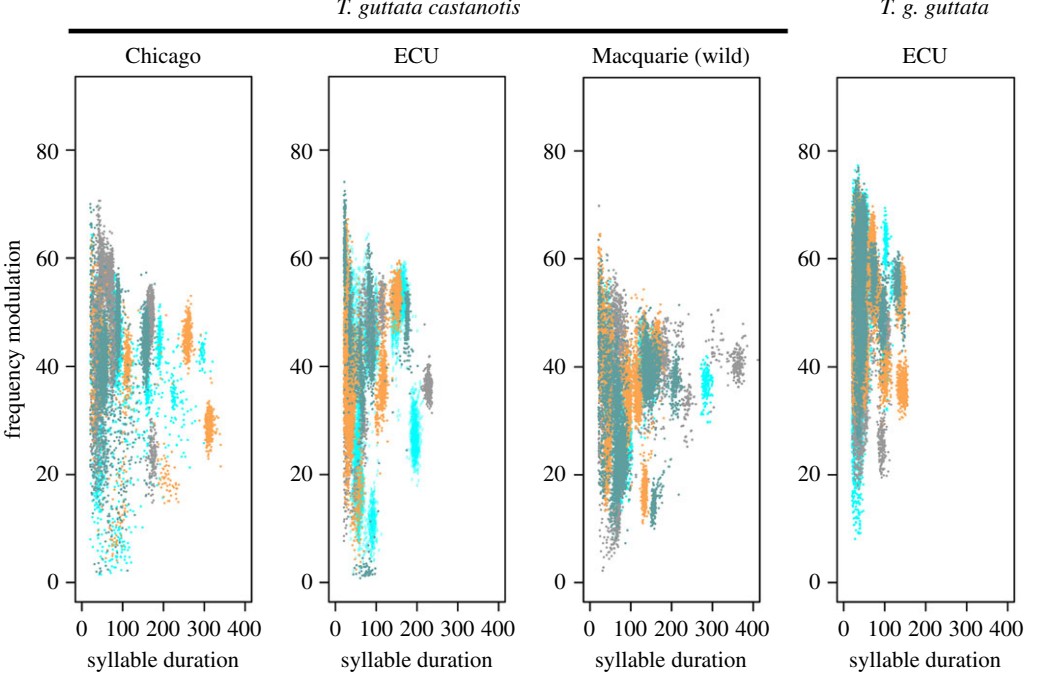

(b)                     song variability in zebra finch populations

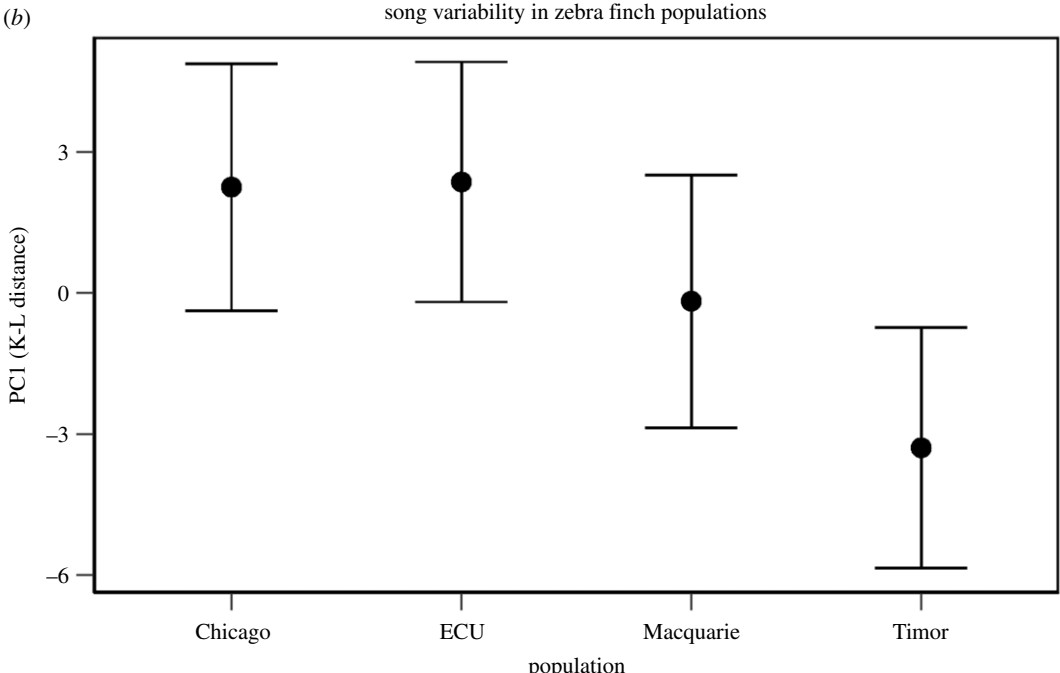

**Figure 2.** (a) Scatterplots of song repertoires of four individuals per population (each colour represents a unique individual). Distinct clusters reflect unique song syllables. Three *TGC* populations are depicted: two domesticated *TGC* populations (one from Chicago and one from ECU), and one wild population of *TGC* from Macquarie. One *TGG* population is depicted from ECU (labelled 'Timor' on graph). (b) Fitted means and confidence intervals for population estimates of K-L distance for principal component 1 for each of the four populations described above reveals population differences in song variability ($\chi^2 = 11.2$, d.f. $= 3$, $p = 0.01$).

heavily used in research. We found that representative colonies of the two subspecies differ in levels of song variability, with *TGG* showing lower variability than its Australian counterpart. While in principle, subspecific differences in variability could be caused by either genetic or cultural evolution, our tutoring experiment suggests that genetic differences contribute to this pattern.

Low population-level song variability within *TGG* in theory could be explained if these birds were more adept at copying their tutors. However, *TGC* and *TGG* did not differ in terms of how closely

they were able to match their Bengalese finch tutors. One challenge in our study was that neither *TGG* nor *TGC* accurately copied the Bengalese finch tutors that were presented to them (figure 1*c*). Although other researchers have successfully trained *TGC* zebra finches to copy Bengalese finches [1,31,32,39,40], we were less successful in this regard. Another possible design would have been to tutor *TGC* with *TGG* and vice versa. Alternatively, synthetic tutor songs could be used. Presenting *TGC* and *TGG* with less divergent templates might facilitate learning and improve our ability to detect differences in song copying should they exist.

Although we cannot rule out differences in copying ability at this point, our tutoring experiment does show that differences in song variability are maintained even after controlling for developmental song exposure. This finding strongly suggests a genetic difference between *TGC* and *TGG* in the mechanisms that generate song variability. Among unmanipulated *TGG* and *TGC*, the former produces syllables that were significantly shorter in duration (figure 2; electronic supplementary material, figure S1). If *TGG* are constrained to sing shorter syllables, these birds would be exploring a smaller acoustic space, explaining the reduced song variability. After cross-fostering, however, we did not observe a difference in mean maximum syllable duration (figure 1*c*), indicating that differences in syllable length alone do not explain the observed difference in song variability. This finding also demonstrates that *TGC*'s use of longer syllables is at least partly determined by experience. Although we controlled for most early song exposure, we note that most of the experimentally tutored birds (13/18) were tutored beginning at post-hatch day 30 rather than from hatch. These birds would have been exposed to other birds from their population early in the sensitive period, and we cannot rule out that such exposure influenced subsequent song variability. Previous work in *TGC*, however, suggests that exposure to tutor song before p30 does not influence tutor song copying [8,41,42].

To better understand the song variability differences observed, we also compared wild-derived and additional domesticated populations within *TGC*. While the wild-derived population was intermediate in song variability between domesticated *TGC* and *TGG* (figure 2), there was only weak evidence of population differences in song variability within *TGC*. Our analyses here required recording of approximately 100 song bouts per bird, a panel of recordings difficult to acquire in the field with wild zebra finches. Thus, we were restricted to sampling a single wild-derived population housed at Macquarie University. Bengalese finches, another domesticated songbird, show more syntactical complexity than their wild ancestors, a pattern attributed to release from species recognition constraints [43]. We might expect that a similar process is taking place in zebra finches and suggest that additional population sampling is warranted to rigorously test this before making generalized conclusions to other populations [44].

Our study reveals population differences in song variability between zebra finch subspecies as represented by captive colonies of each. As auditory experience shapes subsequent processing of acoustic stimuli [14], consideration of these population differences will be important for future studies of vocal behaviour and song recognition. If high song variability is adaptive for individual recognition in large, wild colonies of *TGC* in Australia, have different selective forces acted to shape song structure in the Lesser Sunda Islands? Previous work has suggested reduced sexual selection in island birds relative to their mainland counterparts [45,46] and reduced sexual selection could explain differences in aspects of vocal signals. Following this prediction *TGG* are notably less sexually dimorphic than *TGC* [20]. Previous studies of *TGG*, *TGC* and hybrids were among the first to document the heritability of structural aspects of song production [26] and recent work has continued to reveal genetic contributions to learned vocal communication behaviour [47]. These descriptions of intraspecific variation in song traits, including the song variability itself, will facilitate the understanding of both the proximate and ultimate mechanisms underlying song behaviour.

Ethics. Animal use was conducted with approval from Institutional Animal Care and Use Committees of ECU (AUP D285), the University of Chicago (ACUP72220) and the University of Illinois Urbana-Champaign (UIUC). Animal use at Macquarie University was approved under AEC ARA 2015/017.

Data accessibility. R code and K-L distance measurements are available at https://github.com/chrisbalakrishnan/SongVariability

Authors' contributions. A.L.L. and C.N.B. designed the study, recorded songs, analysed the data and wrote the manuscript. K.M.S. conducted video analyses. S.E.L. conducted song recordings and edited the manuscript. D.F.C. designed the study and edited the manuscript. S.C.G. provided access to wild-derived birds and edited the manuscript.

Competing interests. There are no competing interests.

Funding. This work was funded by East Carolina University and NIH NIGMS 1RC1GM091556-01 (D.F.C.).

Acknowledgements. We would like to thank Ken Soderstrom, Ofer Tchernichovski, Carol Goodwillie, Michael Brewer, Jeff McKinnon, Sue McRae, Mike McCoy and Ariane Peralta for their assistance with this project. Mike McCoy in particular provided extensive help with our introduction to linear mixed-effects modelling.

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
