## [Reviewer comments · Royal Society Open Science]

Review History

RSOS-190273.R0 (Original submission)

Review form: Reviewer 1

Is the manuscript scientifically sound in its present form?

Yes

Are the interpretations and conclusions justified by the results?

Yes

Is the language acceptable?

Yes

Is it clear how to access all supporting data?

Yes

Do you have any ethical concerns with this paper?

No

Have you any concerns about statistical analyses in this paper?

No

Recommendation?

Accept with minor revision (please list in comments)

Comments to the Author(s)

I am very happy with the careful revision that was done in order to incorporate all my previous comments; so I have only a few rather minor comments remaining.

I very much appreciate the cautious interpretation that is made throughout the Discussion section. Would it be possible to add another 1-2 sentences addressing the problem of uncertainty whether the current findings will generalize to other populations (possibly referring to the Byers 2011 Anim Behav paper that I mentioned previously for a more detailed discussion of the problem)?

Throughout the Results section: Please ensure that for all non-significant results we not only get to see a p-value but also some measure of the direction of the trend which would facilitate use in a meta-analysis (unless $p=1$). For instance one could report mean values for each of the two subspecies, which would clarify the direction of the trend without having to call this a 'trend' (because some readers are opposed to using the term 'trend' for effects that come with clearly non-significant p-values).

Minor comments line by line:

1. Line 87: "wild": do you mean "wild and recently wild-derived"?
2. Line 176: "preened": I have never noticed preening as part of the courtship dance. I guess that you are not referring to the typical preening behavior where male and female sit in body contact often for several minutes and allopreen.
3. Line 209: "unique": The wording is unclear to me. Do you mean that thresholds were manually chosen once for every individual (thereby different thresholds for each individual)?
4. Line 247: "generalized": This term is widely used for non-Gaussian models (e.g. Poisson), but I guess you fitted a Gaussian model.
5. Lines 283-285 and elsewhere: If I remember PCA correctly, then the 'loadings' are the correlations ($-1 < r < +1$) between the original variables and the PC. So for instance if all variables would have the same strong positive loading of $r=+0.90$, then the mean r-squared would be 0.81, meaning that PC1 explains 81% of the total variance. The numbers that are given here (loadings of 0.23 to 0.3) do not seem to fit with the notion that PC1 explains 86.5% of the variance. Can you clarify where this discrepancy with my expectations comes from? Is it that the loadings are calculated differently, or is it that PC1 explains 86.5%, PC2 explains 13.5%, and PC3 to PC13 all get rejected due to their eigenvalues being <1 and therefore are not entered in the calculation? If the latter is true, then I find this highly misleading because then it would not be the proportion of total variance "in the song data".
6. Line 299: Change to "In a group of 6 TGC and 6 TGG finches..."
7. Line 301: Report two medians or means rather than just one.

Review form: Reviewer 2

Is the manuscript scientifically sound in its present form?

Yes

Are the interpretations and conclusions justified by the results?

Yes

Is the language acceptable?

Yes

Is it clear how to access all supporting data?

Yes

Do you have any ethical concerns with this paper?

No

Have you any concerns about statistical analyses in this paper?

I do not feel qualified to assess the statistics

Recommendation?

Accept with minor revision (please list in comments)

Comments to the Author(s)

The paper compares subspecies of zebra finch in how variable songs are within population. Both domesticated and a wild-derived population of the subspecies *Taeniopygia guttata castanotis* (TGC) were analyzed and the much more recently captivated *taeniopygia guttata guttata* (TGG, Timor) finches. In domesticated TGC, song diversity was highest, whereas in TGG diversity was lowest and the wild-derived TGC population was intermediate.

Birds of the two subspecies were also cross-fostered with Bengalese finches, resulting in a similar pattern: TGC more variable than TGG. This indicates that the origin in variation likely has a genetic component.

These findings are highly relevant as it gives insight in the evolution of vocalizations, the relation between genetic and cultural evolution and their effect on learning mechanisms.

I have seen this manuscript before and my earlier concerns have been adequately dealt with: 1) the potential unequal amount of directed and undirected song in the populations have been measured and turns out to be equal. 2) The statistics have been adjusted and p-values now match more what I would expect based on the graphs.

All other questions have been answered satisfactory as well.

I only have some minor questions and comments.

-Statistics: I appreciate the work done to improve the statistics. I don't know enough about bayesian statistics to see if the current statistics are appropriate and I don't understand the singularity issue. I'm also not sure if the lmer nesting is implemented correctly. I would expect something like $1 | \text{bird}/\text{pair}$ (where pair is each (average) comparison between 2 individuals). Then you would need one extra column with individual label in added to the file now on github if I'm correct. I'm not sure if the current way of nesting you use could also be correct so please verify with a statistician.

That being said, I think the p-values make sense given the graphs so I don't expect very different results.

-Is it possible a data point is missing from the data file? I thought there should be 1 or more comparison in Chicago?

-Reviewer 1 asked a question about syllable duration, which reminded me that actually previous cross-fostering experiments with Bengalese showed gap duration is one of the features that seem to be experience independent as well as phrase length (see Araki et al., 2016, clayton et al., 89). How do your results relate to those studies?

Decision letter (RSOS-190273.R0)

04-Apr-2019

Dear Dr Lansverk

On behalf of the Editors, I am pleased to inform you that your Manuscript RSOS-190273 entitled "The variability of song variability in zebra finch (*Taeniopygia guttata*) populations" has been accepted for publication in Royal Society Open Science subject to minor revision in accordance with the referee suggestions. Please find the referees' comments at the end of this email.

The reviewers and handling editors have recommended publication, but also suggest some minor revisions to your manuscript. Therefore, I invite you to respond to the comments and revise your manuscript.

- Ethics statement

- Data accessibility

If you wish to submit your supporting data or code to Dryad (<http://datadryad.org/>), or modify your current submission to dryad, please use the following link:
<http://datadryad.org/submit?journalID=RSOS&manu=RSOS-190273>

- Competing interests

- Authors' contributions

All submissions, other than those with a single author, must include an Authors' Contributions section which individually lists the specific contribution of each author. The list of Authors

should meet all of the following criteria; 1) substantial contributions to conception and design, or acquisition of data, or analysis and interpretation of data; 2) drafting the article or revising it critically for important intellectual content; and 3) final approval of the version to be published.

- Acknowledgements

- Funding statement

Because the schedule for publication is very tight, it is a condition of publication that you submit the revised version of your manuscript before 13-Apr-2019. Please note that the revision deadline will expire at 00.00am on this date. If you do not think you will be able to meet this date please let me know immediately.

- 1) A text file of the manuscript (tex, txt, rtf, docx or doc), references, tables (including captions) and figure captions. Do not upload a PDF as your "Main Document";

- 2) A separate electronic file of each figure (EPS or print-quality PDF preferred (either format should be produced directly from original creation package), or original software format);
- 3) Included a 100 word media summary of your paper when requested at submission. Please ensure you have entered correct contact details (email, institution and telephone) in your user account;
- 4) Included the raw data to support the claims made in your paper. You can either include your data as electronic supplementary material or upload to a repository and include the relevant doi within your manuscript. Make sure it is clear in your data accessibility statement how the data can be accessed;
- 5) All supplementary materials accompanying an accepted article will be treated as in their final form. Note that the Royal Society will neither edit nor typeset supplementary material and it will be hosted as provided. Please ensure that the supplementary material includes the paper details where possible (authors, article title, journal name).

on behalf of Prof Kevin Padian (Subject Editor)
openscience@royalsociety.org

Associate Editor Comments to Author:

Overall, the reviewers are in favour of publication; however, they both point out a number of items that should be addressed to improve an already promising manuscript. Please ensure you respond to these suggestions, and incorporate them into the manuscript before resubmitting. Good luck!

Reviewer comments to Author:

Reviewer: 1

Comments to the Author(s)

I am very happy with the careful revision that was done in order to incorporate all my previous comments; so I have only a few rather minor comments remaining.

I very much appreciate the cautious interpretation that is made throughout the Discussion section. Would it be possible to add another 1-2 sentences addressing the problem of uncertainty whether the current findings will generalize to other populations (possibly referring to the Byers 2011 Anim Behav paper that I mentioned previously for a more detailed discussion of the problem)?

Throughout the Results section: Please ensure that for all non-significant results we not only get to see a p-value but also some measure of the direction of the trend which would facilitate use in a meta-analysis (unless $p=1$). For instance one could report mean values for each of the two subspecies, which would clarify the direction of the trend without having to call this a 'trend' (because some readers are opposed to using the term 'trend' for effects that come with clearly non-significant p-values).

Minor comments line by line:

1. Line 87: "wild": do you mean "wild and recently wild-derived"?
2. Line 176: "preened": I have never noticed preening as part of the courtship dance. I guess that you are not referring to the typical preening behavior where male and female sit in body contact often for several minutes and allopreen.
3. Line 209: "unique": The wording is unclear to me. Do you mean that thresholds were manually chosen once for every individual (thereby different thresholds for each individual)?
4. Line 247: "generalized": This term is widely used for non-Gaussian models (e.g. Poisson), but I guess you fitted a Gaussian model.
5. Lines 283-285 and elsewhere: If I remember PCA correctly, then the 'loadings' are the correlations ($-1 < r < +1$) between the original variables and the PC. So for instance if all variables would have the same strong positive loading of $r=+0.90$, then the mean r-squared would be 0.81, meaning that PC1 explains 81% of the total variance. The numbers that are given here (loadings of 0.23 to 0.3) do not seem to fit with the notion that PC1 explains 86.5% of the variance. Can you clarify where this discrepancy with my expectations comes from? Is it that the loadings are calculated differently, or is it that PC1 explains 86.5%, PC2 explains 13.5%, and PC3 to PC13 all get rejected due to their eigenvalues being <1 and therefore are not entered in the calculation? If the latter is true, then I find this highly misleading because then it would not be the proportion of total variance "in the song data".
6. Line 299: Change to "In a group of 6 TGC and 6 TGG finches..."
7. Line 301: Report two medians or means rather than just one.

Reviewer: 2

Comments to the Author(s)

The paper compares subspecies of zebra finch in how variable songs are within population. Both domesticated and a wild-derived population of the subspecies *Taeniopygia guttata castanotis* (TGC) were analyzed and the much more recently captivated *taeniopygia guttata guttata* (TGG, Timor) finches. In domesticated TGC, song diversity was highest, whereas in TGG diversity was lowest and the wild-derived TGC population was intermediate.

Birds of the two subspecies were also cross-fostered with Bengalese finches, resulting in a similar pattern: TGC more variable than TGG. This indicates that the origin in variation likely has a genetic component.

These findings are highly relevant as it gives insight in the evolution of vocalizations, the relation between genetic and cultural evolution and their effect on learning mechanisms.

I have seen this manuscript before and my earlier concerns have been adequately dealt with: 1) the potential unequal amount of directed and undirected song in the populations have been measured and turns out to be equal. 2) The statistics have been adjusted and p-values now match more what I would expect based on the graphs.

All other questions have been answered satisfactory as well.

I only have some minor questions and comments.

-Statistics: I appreciate the work done to improve the statistics. I don't know enough about bayesian statistics to see if the current statistics are appropriate and I don't understand the singularity issue. I'm also not sure if the lmer nesting is implemented correctly. I would expect something like 1 | bird/pair (where pair is each (average) comparison between 2 individuals). Then you would need one extra column with individual label in added to the file now on github if I'm correct. I'm not sure if the current way of nesting you use could also be correct so please verify with a statistician.

That being said, I think the p-values make sense given the graphs so I don't expect very different results.

-Is it possible a data point is missing from the data file? I thought there should be 1 or more comparison in Chicago?

-Reviewer 1 asked a question about syllable duration, which reminded me that actually previous cross-fostering experiments with Bengalese showed gap duration is one of the features that seem to be experience independent as well as phrase length (see Araki et al., 2016, clayton et al., 89). How do your results relate to those studies?

Author's Response to Decision Letter for (RSOS-190273.R0)

See Appendix A.

Decision letter (RSOS-190273.R1)

12-Apr-2019

Dear Dr Lansverk,

I am pleased to inform you that your manuscript entitled "The variability of song variability in zebra finch (*Taeniopygia guttata*) populations" is now accepted for publication in Royal Society Open Science.

on behalf of Prof Kevin Padian (Subject Editor)
openscience@royalsociety.org

Follow Royal Society Publishing on Twitter: [@RSocPublishing](https://twitter.com/RSocPublishing)
Follow Royal Society Publishing on Facebook:
<https://www.facebook.com/RoyalSocietyPublishing.FanPage/>
Read Royal Society Publishing's blog: <https://blogs.royalsociety.org/publishing/>

Appendix A

Dear Reviewers and Editor,

We thank you for your thoughtful comments on our MS. We have corrected the minor issues identified by the reviewers. Please see detailed responses below.

Reviewer comments to Author:

Reviewer: 1

Comments to the Author(s)

I am very happy with the careful revision that was done in order to incorporate all my previous comments; so I have only a few rather minor comments remaining.

I very much appreciate the cautious interpretation that is made throughout the Discussion section. Would it be possible to add another 1-2 sentences addressing the problem of uncertainty whether the current findings will generalize to other populations (possibly referring to the Byers 2011 Anim Behav paper that I mentioned previously for a more detailed discussion of the problem)?

We expanded on a sentence we already had in the discussion on line 399-400 to further emphasize the need for including additional populations before generalizing conclusions (line 404-405) and included the Byers 2011 reference (line 566-567)

Throughout the Results section: Please ensure that for all non-significant results we not only get to see a p-value but also some measure of the direction of the trend which would facilitate use in a meta-analysis (unless $p=1$). For instance one could report mean values for each of the two subspecies, which would clarify the direction of the trend without having to call this a 'trend' (because some readers are opposed to using the term 'trend' for effects that come with clearly non-significant p-values).

I added in means for the two populations (line 304-305). For everything else, to enhance reproducibility and facilitate meta-analysis, all the code and data are in the GitHub, and the processed html is provided in the supplement.

Minor comments line by line:

1. Line 87: "wild": do you mean "wild and recently wild-derived"?

Added in "wild-derived" (line 87)

2. Line 176: "preened": I have never noticed preening as part of the courtship dance. I guess that you are not referring to the typical preening behavior where male and female sit in body contact often for several minutes and allopreen.

This was more of an ephemeral behavior than is suggested by Reviewer 1's definition of allopreening. Occasionally, immediately after a directed singing bout, we noticed that a male would preen the female's neck for 1-3 seconds. Changed the word "preen" to "groom" so it clarifies (line 176)

3. Line 209: "unique": The wording is unclear to me. Do you mean that thresholds were manually chosen once for every individual (thereby different thresholds for each individual)?

Yes, each individual has distinct syllables so thresholds were set for each bird. Changed the wording to “We used the Feature Batch function in SAP to parse the motifs into syllables by manually setting segmentation values for amplitude, mean frequency and continuity once for each individual and then parsing all recordings” (line 209)

4. Line 247: “generalized”: This term is widely used for non-Gaussian models (e.g. Poisson), but I guess you fitted a Gaussian model.

Removed the word generalized (line 249)

5. Lines 283-285 and elsewhere: If I remember PCA correctly, then the ‘loadings’ are the correlations ($-1 < r < +1$) between the original variables and the PC. So for instance if all variables would have the same strong positive loading of $r = +0.90$, then the mean r -squared would be 0.81, meaning that PC1 explains 81% of the total variance. The numbers that are given here (loadings of 0.23 to 0.3) do not seem to fit with the notion that PC1 explains 86.5% of the variance. Can you clarify where this discrepancy with my expectations comes from? Is it that the loadings are calculated differently, or is it that PC1 explains 86.5%, PC2 explains 13.5%, and PC3 to PC13 all get rejected due to their eigenvalues being < 1 and therefore are not entered in the calculation? If the latter is true, then I find this highly misleading because then it would not be the proportion of total variance “in the song data”.

There is variation in the use of the term “loadings”. Loadings generated by the `prcomp` function in R represent the eigenvectors of the covariance matrix (principal directions, or principal axes). To get traditional loadings, these eigenvectors can be multiplied by the square root of the explained variance. Since we’ve specified that we are using “loadings” from the `prcomp` function, and have shared the associated code, we’ve opted to leave this as is. We’ve specified eigenvectors parenthetically in the text.

6. Line 299: Change to “In a group of 6 TGC and 6 TGG finches...”

Changed (line 302).

7. Line 301: Report two medians or means rather than just one.

Added in the means for the two populations separately (line 304-305).

Reviewer: 2

Comments to the Author(s)

The paper compares subspecies of zebra finch in how variable songs are within population. Both domesticated and a wild-derived population of the subspecies *Taeniopygia guttata castanotis* (TGC) were analyzed and the much more recently captivated *taeniopygia guttata guttata* (TGG, Timor) finches. In domesticated TGC, song diversity was highest, whereas in TGG diversity was lowest and the wild-derived TGC population was intermediate.

Birds of the two subspecies were also cross-fostered with Bengalese finches, resulting in a similar pattern: TGC more variable than TGG. This indicates that the origin in variation likely has a genetic component.

These findings are highly relevant as it gives insight in the evolution of vocalizations, the relation between genetic and cultural evolution and their effect on learning mechanisms.

I have seen this manuscript before and my earlier concerns have been adequately dealt

with: 1) the potential unequal amount of directed and undirected song in the populations have been measured and turns out to be equal. 2) The statistics have been adjusted and p-values now match more what I would expect based on the graphs. All other questions have been answered satisfactory as well.

I only have some minor questions and comments.

-Statistics: I appreciate the work done to improve the statistics. I don't know enough about bayesian statistics to see if the current statistics are appropriate and I don't understand the singularity issue. I'm also not sure if the lmer nesting is implemented correctly. I would expect something like 1|bird/pair (where pair is each (average) comparison between 2 individuals). Then you would need one extra column with individual label in added to the file now on github if I'm correct. I'm not sure if the current way of nesting you use could also be correct so please verify with a statistician. That being said, I think the p-values make sense given the graphs so I don't expect very different results.

We verified with a statistician (Michael McCoy) when running the stats and are confident we've done these correctly.

-Is it possible a data point is missing from the data file? I thought there should be 1 or more comparison in Chicago?

I think this is referring to confusion from the language in the Figure 2 label where it says "Three TGC populations are depicted: two domesticated TGC populations from Chicago and ECU, and one wild population of TGC from Macquarie"? I changed the language slightly. It now reads "Three TGC populations are depicted: two domesticated TGC populations (one from Chicago and one from ECU), and one wild population of TGC from Macquarie." (line 436-437)

-Reviewer 1 asked a question about syllable duration, which reminded me that actually previous cross-fostering experiments with Bengalese showed gap duration is one of the features that seem to be experience independent as well as phrase length (see Araki et al., 2016, clayton et al., 89). How do your results relate to those studies?

Unfortunately, we did not measure gap duration and going is beyond the scope of what we are able to do at this stage.